



# Impact of the current feedback on kinetic energy over the North-East Atlantic from a coupled ocean / atmospheric boundary layer model.

Théo Brivoal[1,2], Guillaume Samson[1], Hervé Giordani[2], Romain Bourdallé-Badie[1], Florian Lemarié[3], Gurvan Madec[4,3]

[1]Mercator Océan, Ramonville-Saint-Agne, 31520, France
[2]Université de Toulouse, Centre National de Recherches Météorologique (CNRM), Météo-France, CNRS, Toulouse, France
[3]Univ. Grenoble Alpes, Inria, CNRS, Grenoble INP, LJK, 38000 Grenoble, France
[4]Sorbonne Universités, CNRS, UPMC, IRD, MNHN, LOCEAN, Paris, France

*Correspondence to*: Théo Brivoal (theo.brivoal@mercator-ocean.fr)

**Abstract.**

A one-dimensional Atmospheric Boundary Layer (ABL1D) is coupled with the NEMO ocean model and implemented over the Iberian–Biscay–Ireland (IBI) area at 1/36° resolution to investigate the retroactions between the surface currents and the atmosphere, namely the Current FeedBack (CFB) in this region of low mesoscale activity. The ABL1D-NEMO coupled model
is forced by a large-scale atmospheric reanalysis (ERA-Interim) and integrated over the period 2016-2017.

The mechanisms of eddy kinetic energy damping and ocean upper-layers re-energization are realistically simulated, meaning that the CFB is properly represented by the model. In particular, the dynamical coupling coefficients between the curls of surface stress / wind and current are in agreement with the literature.

The effects of CFB on the kinetic energy (KE) are then investigated through a KE budget. We show that the KE decrease
induced by the CFB is significant down to 1500m. Near the surface (0 – 300m), most of the KE decrease can be explained by a reduction of the surface wind work by 4%. At depth (300 – 2000m), the CFB induce a reduction of the pressure work (i.e: the PE to KE conversion) associated with a reduction of KE which is significant down to 1500m. We show that this reduction of KE at depth can be explained by CFB-induced Ekman pumping above eddies that weakens the mesoscale activity and this over the whole water column.






## 1 Introduction


Global ocean operational systems have recently initiated a transition towards mesoscale-resolving resolutions (scales of ten to hundred kilometres). The advent of satellite scatterometers measurement of the surface wind speed has provided evidence that the signature of mesoscale activities can be clearly seen in the wind stress fields. At mesoscale, a positive correlation is found between Sea Surface Temperature (SST) and surface wind stress ( Wentz et al., 2000; Chelton and Wentz, 2005), suggesting

that the ocean is driving the atmosphere, whereas the atmosphere drives the ocean response at larger scale (Liu et al., 1994). At mesoscale, the atmospheric adjustment to the current and SST anomalies associated with the ocean eddy field (Small et al., 2008) sets up through the surface heat and momentum fluxes (e.g: Chelton et al., 2001; Bourras et al., 2004; Samelson et al., 2006; Chelton and Xie, 2010; Frenger et al., 2013; O'Neill et al., 2012; Renault et al., 2016).

The surface wind-stress is often computed as a function of the absolute wind $U_a$, which amounts to consider that the ocean is

motionless, whereas it should be function of the relative wind $(U_a - U_o)$ (Bye, 1986; Dewar and Flierl, 1987). The induced surface current wind-stress change is called *"Stress Current FeedBack"* (CFB_stress). On average and at a global scale, the CFB_stress has moderate effects since it reduces the wind stress and heat fluxes by $1 - 7$ % (Dawe and Thompson, 2006; Duhaut and Straub, 2006) and the wind-work input by $20 - 35$ % (Dawe and Thompson, 2006; Duhaut and Straub, 2006) in forced simulations mode.

At large scale, the reduction of wind-work by currents causes a slowdown in the general oceanic circulation (e.g: Pacanowski, 1987; Luo et al., 2005; Renault et al., 2016a, 2017), whereas at mesoscale it induces of loss of kinetic energy towards the atmosphere through negative eddy wind-work because of opposed wind stress curl and current vorticity anomalies in eddies ( Rooth and Xie, 1992; Renault et al., 2016b; Oerder et al., 2018). In forced simulations mode, this may lead to a reduction of the Eddy Kinetic Energy (EKE) from 10 to 50% depending on regions (e.g : Duhaut and Straub, 2006; Zhai and Greatbatch,

2007; Yu and Metzger, 2019) . Besides, the current – stress interaction can also modify the eddies structure and thus the Ekman pumping through the current feedback (Gaube et al., 2014; Seo et al., 2015; Oerder et al., 2018) with the result of increasing the stratification near the bottom of the mixed layer by up to 10% (Song et al., 2020).







In forced simulations mode, the wind cannot adjust to the change in surface stress caused by the currents. The difference of inertia between the ocean and the atmosphere may introduce significative error in the prescribed atmospheric field because the
atmosphere reacts much faster than the ocean.

In coupled models, the wind can adjust and a positive linear relationship is found between the surface wind and current curls (Renault et al., 2016b), which partially re-energizes the ocean, leading to more realistic EKE levels (Renault et al., 2016b, 2019; Jullien et al., 2020). This atmospheric response that re-energizes the ocean is called *"Wind Current FeedBack"* (CFB_wind). In coupled mode, the *"Total Current FeedBack"* (CFB_tot) is the sum of CFB_stress and CFB_wind, which are
the oceanic and atmospheric retroaction loops, respectively. The kinetic energy damping caused by the current feedback exhibits a strong regional variability and is related to both mean wind intensity and EKE (Jullien et al., 2020). A proper representation of the current feedback improves the realism of simulated structures such as the Gulf-Stream separation and post-separation at Cape Hatteras (Renault et al., 2016a) or the position of the Agulhas current retroflection (Renault et al., 2017a), therefore influencing the regional and global climate systems. However, the effect of CFB on less dynamical regions
remains poorly documented.

Correct current feedback representation (i.e. representing CFB_tot instead of CFB_stress only) necessitates both an ocean models resolving mesoscale fields and an atmosphere model at the ocean model effective resolution (Jullien et al., 2020), but the computational cost of such configurations remains limiting for long term analysis and operational perspectives.

Alternative approaches to mimic the full current feedback are to parameterize the atmospheric retroaction of the current
feedback in forced models (Renault et al., 2017b), or to use simplified atmospheric models, which aim at resolving the wind response to the surface properties and thus representing the atmospheric loop of the current feedback mechanism. In this study, we use a one-dimensional atmospheric boundary layer model (ABL1D; Lemarie et al., 2020) to simulate the temperature, the humidity and the wind changes generated by turbulent processes at play in the planetary boundary layer. The ABL1D model was validated on idealized test cases and was recently implemented within the NEMO ocean model (Lemarie et al., 2020)
The present paper aims at presenting the validity of the ABL1D model in the North-eastern Atlantic, which is a region of low mesoscale activity, and to investigate the current feedback effects on the circulation and kinetic energy budget in this region. Coupled and forced twin experiments with absolute and relative winds are performed and compared over the two-year period 2016-2017 to validate the ABL1D winds with observations and the current – wind-stress coupling coefficients (Renault et al., 2016a). The associated changes of behaviour are discussed using an online KE budget.





## 2 Numerical models

### 2.1 ABL1D

The ABL1D model is a single column model that reproduces the turbulent processes within the Planetary Boundary Layer (PBL). The single-column formulation is valid under the following hypothesis: i) Horizontal homogeneity ($\partial x = \partial y = 0$), ii)

the atmosphere within the ABL is transparent and iii) vertical motions are neglected (w=0). However, these approximations can be compensated by nudging the model prognostic variables towards the large-scale ones. The equations set of the ABL1D model is written as follows (Lemarié et al., 2020):

$$\partial_t \overrightarrow{U_h} = -f\mathbf{k} \times (\mathbf{U_h} - \mathbf{U_{geo}}) + \partial_z(K_m \partial_z \mathbf{U_h}) \tag{1}$$

$$\partial_t \theta = \partial_z(K_s \partial_z \theta) + \lambda_s(\theta - \theta_{LS}) \tag{2}$$

$$\partial_t q = \partial_z(K_s \partial_z q) + \lambda_s(q - q_{LS}) \tag{3}$$

Where $\mathbf{U_h}$ is the horizontal wind vector, $q$ is the specific humidity and $\theta$ is the potential temperature, $K_m$ and $K_s$ are the turbulent viscosity and diffusivity coefficients. $\mathbf{U_{geo}}$ is the geostrophic wind vector, $\theta_{LS}$ and $q_{LS}$ are the large-scale variables potential

temperature and humidity, respectively, and $\lambda_s$ is a nudging term to relax the prognostic variables toward the large-scale atmospheric state. The large-scale variables are typically prescribed from an Atmospheric Global Circulation Model (AGCM). Here, the ERA-interim atmospheric reanalysis (Dee et al., 2011) dataset is used to force both the ABL1D model (for $u$, $v$, $\theta$ and $q$) and the ocean model in forced configurations.

For the dynamics, the assumption is that the ageostrophic part of the dynamics is only related to the turbulence within the PBL

and is consequently resolved by ABL1D. The large-scale motions are approximated as geostrophic and the geostrophic wind $U_{geo}$ is computed under the hydrostatic equilibrium hypothesis from the Mean Sea Level Pressure (*MSLP*) field obtained from ECMWF datasets through a preprocessing tool that takes into input the forcing fields $u$, $v$, $\theta$ and $q$ and *MSLP* (see Lemarié et al. 2020 for more details).

ABL1D computes $u$, $v$, $\theta$ and $q$ evolutions between the surface and 2000 m. Other forcing fields such as the radiative fluxes,

precipitation and surface pressure are prescribed to the ocean surface and remain unchanged compared to the usual forced mode. The differences between ABL1D winds and ERA-interim large-scale winds can originate from different effects which are changes in the vertical shear due to surface stress modification by the currents or SST-induced drag coefficient changes, SST-induced buoyancy changes, differences in the turbulence parameterization (CBR, Cuxart et al., 2000) and unresolved processes in ABL1D.




The tracers are nudged towards large-scale data by using a relaxation coefficient which varies with altitude. Although the ABL1D is able to reproduce the SST feedback on the buoyancy in the PBL, we chose a relaxation coefficient for the tracers equal to the model time step ($\Delta t$=150s). As a consequence, the tracers are not modified by the ABL1D in order to focus on the vertical wind shear changes in the PBL induced by surface conditions. In the frame of this experimental setup, ABL1D winds are only modified by surface conditions though current-induced surface wind-stress changes or drag coefficient-induced SST

changes. Note that the turbulent heat fluxes also depend on the SST and the wind speed and are consequently modified by the ABL1D model.

To compute the vertical eddy viscosity and diffusivity coefficients, $K_m$ and $K_s$, the turbulent closure scheme implemented in ABL1D is derived from the CBR-1D scheme of (Cuxart et al., 2000).The viscosity and diffusivity coefficients are computed from the total kinetic energy $e$ and the mixing length L as: $K_s = \frac{2}{3}\frac{L}{C_s}\phi_z\sqrt{e}$, $K_m = \frac{4}{15}\frac{L}{C_m}\sqrt{e}$ .

Where $C_e = 0.4$ and $C_m = 4$ and the stability function $\phi_z$ is defined as:

$$\phi_z(z) = \left(1 + \left\{\frac{C_1 L^2 N^2}{e}, -0,5455\right\}\right)^{-1} \qquad (4)$$

The mixing length L is computed using an adaptation of the NEMO algorithm (Sec. 10.1.3 in Madec, 2012) in which the mixing length is computed from the Deardorff (1980) scheme, where L depends on the Brünt-Vaisala frequency $N^2$ and the total kinetic energy and then limited such that they stay smaller than the variations of height (Lemarié et. al. 2020).

The vertical grid of the model has 50 vertical levels. From the surface to 200m, the grid resolution is about 20 m, then increases

monotonically from 20m to 90m at the top (2000m). The ABL1D model has the same time step (Dt=150s) and horizontal grid-mesh ($\Delta$x=1/36° ≈ 3 km) as the NEMO ocean model. As the model is directly implemented within the NEMO model, no model coupler is required for coupling the ABL1D with NEMO.





## 2.2 Ocean model (NEMO 3.6)


The ocean component, based on the primitive equations, is the Nucleus for European Modelling of the Ocean (NEMO, Madec, 2009) in its version 3.6. NEMO is implemented over the IBI (Iberian, Biscay, Ireland, Maraldi et al., 2013) area at 1/36° horizontal resolution. The horizontal grid is a subdomain of the tripolar grid ORCA (Madec, 2009) on the North-East Atlantic shelf. The model has 50 vertical levels with a stretching ranging from 1m in the first 10 meters to more than 400m in the deep
ocean. The baroclinic time step is 150 s.

One day averages of the global analysis PSY4V3R1 at 1/12° resolution are used as lateral boundary conditions of the IBI domain. The forcing associated with tides is represented by 11 tidal harmonics (M2, S2, N2, K1, O1, Q1, M4, K2, P1, Mf, Mm) provided by FES2004 (Lyard et al., 2006) and TPXO7.1 (Egbert and Y. Erofeeva, 2002) data sets. Freshwater runoff is implemented as lateral boundary conditions for 33 rivers which is based on a blended product derived from observations
(PREVIMER project), simulated data (SMHI E-HYPE hydrological model) and climatology (Bourdallé-Badie and Treguier, 2006).

The vertical mixing is parameterized by a k-ε model implemented in the generic form proposed by Umlauf and Burchard, (2003). At the air-sea interface, the fluxes are computed using the ECMWF bulk algorithm (Beljaars, 1995) from the AEROBULK package (Brodeau et al., 2017, https://github.com/brodeau/aerobulk). This regional configuration details and
validation can be found in Maraldi et. al. (2013).






### 2.3 Simulation design

Two sets of twin experiments in forced and coupled mode have been performed over the period from the 19[th] December 2015
to the 31[st] December 2017. The two sets differ in the way the surface wind-stress is computed, either using absolute winds or
relative winds. All simulations are forced with the same ERA-interim fields which have a temporal resolution of 6h and a
horizontal resolution of ¾°.

In the spirit of Renault et al., (2016b) the behaviour of the ABL1D model regarding the CFB mechanism is investigated in two
coupled (ABL1D-NEMO) simulations, the first one with surface current (ABL_REL) and the second one with no surface
current (ABL_ABS) in the surface wind-stress formulation. Two additional reference ocean simulations forced directly by
ERA-interim data are also performed with absolute wind (FRC_ABS) and relative wind (FRC_REL). A summary of these
four simulations setups can be found in Table 1.

Differences between FRC_ABS and FRC_REL are only caused by the CFB_stress mechanism, whereas the differences
between ABL_ABS and ABL_REL are caused by the total CFB (CFB_tot). The wind differences between ABL_ABS and
FRC_ABS are attributable to the turbulence parameterizations used in ABL1D and to the ERA-i data assimilation procedure
for one part and for another part to non-resolved dynamical processes in ABL1D. The wind difference between ABL_REL
and FRC_REL is attributable to the same processes, with the additional impact of the CFB_wind effect and the SST-induced
drag coefficient changes. However, the drag coefficient change generally has a weak impact on the surface wind-stress (e.g:
O'Neill et al., 2012). All simulations were initialized from the same ABL ABS simulation which were performed over the
period from the 1[st] January 2015 to 19[th] December 2015. This 11-month "spinup" simulation were initialized from the IBI
MFC ocean analysis and forecast system at 1/36°, where sea level, SST and in-situ T-S profiles were assimilated. When
compared with forced simulation, the computational cost of ABL1D - NEMO simulations is increased by 12%.

| Simulation | Atmospheric conditions | Absolute / Relative winds | Current feedback |
|---|---|---|---|
| ABL_ABS | ABL1D – ERA-interim | Absolute | No |
| ABL_REL | ABL1D – ERA-Interim | Relative | CFB_stress + CFB_wind |
| FRC_ABS | ERA-interim | Absolute | No |
| FRC_REL | ERA-Interim | Relative | CFB_stress |

**Table 1: Summary of the differences between experiments**




# 3 Results

## 3.1 Validation

ABL1D winds are evaluated against L4 ASCAT reprocessed satellite dataset (Bentamy et al., 2012, http://products.cersat.fr/details/?id=CER_WND_GLO_1D_025_ASCAT). This dataset is a blended product from regridded
observations of two scatterometers ASCAT-A and ASCAT-B (onboard the METOP-A and METOP-B satellites) and ERA-interim reanalysis. ASCAT winds are daily products with a spatial resolution of 0.25°. Details related to data, methodology, and product accuracy issues can be found in Bentamy et al., (2012).

In order to compare the simulated winds with various observation products, model equivalent-neutral winds $u_n$ are computed as follows (Hersbach, 2010):

$$u_n(z) = \frac{u_*}{\kappa} log\ (1 + \frac{z}{z_0})$$  (5)

Where $u_*$ is the friction velocity in m/s, $\kappa = 0.41$ is the Von Karman constant, z is the wind height (taken equal to 10 m here) and $z_0$ is the roughness length in m. Stress-equivalent wind $u_s$ is then computed from the equivalent-neutral winds following de Kloe et al., (2017):

$$u_s = \sqrt{\frac{\rho_a}{\rho_{a_0}}} u_n$$  (6)

Where $\rho_a$ is the air density, $\rho_{a_0} = 1,22$ kg/m³. Finally, $u_s$ is regridded with a bilinear interpolation on the ASCAT grid.

Figure 1.a presents the binned scatterplots of ABL1D and ERA-I surface daily stress-equivalent wind versus ASCAT wind.
Data near the coast (< 150 km) have been masked to avoid coastal contamination and accuracy issues. The mean difference between ABL_ABS and ABL_REL winds is small (< 0.1 m/s) and the Root Mean Square Difference (RMSD) between the two simulations is equal to 0.08 m/s as the currents are much smaller than the total winds and the remapping towards ASCAT grid may damp the current-induced wind anomalies.

When compared with ASCAT, ABL simulated stress-equivalent winds are globally stronger with a RMSE of 1.98 m/s and a
mean difference with ASCAT of +0.29 m/s (+3.6%). On the other hand, ERA-Interim stress-equivalent winds are weaker, with a standard deviation of 1.43 m/s and a mean difference with ASCAT of -0.27 m/s (-3.2%). Discrepancies between ABL, ERA-interim and observations increase with wind intensity. Globally, ABL1D produces realistic wind for magnitudes weaker than 15 m/s and overestimates stronger winds. ABL wind differences with ASCAT increase from 2 to 4 m/s beyond 16 m/s, but the occurrence of these strong winds is very weak over the IBI domain. Such overestimation of strong winds could be
related to the choice of the bulk parameterization or to the effect of waves on drag coefficient which is not represented in our experiments.

Although winds are stronger in winter than in summer over the IBI area (Fig. 2.a), no significant seasonal cycle is observed on the wind speed bias (Fig. 2.b). The mean bias is ranging from -0.3 m/s to +0.7 m/s (-4% to + 9%) for ABL1D and -0.5 m/s





to +0.1 m/s (-6% to + 1%) for ERA-interim. Again, differences between ABL_ABS and ABL_REL are very small in Fig. 2 as the winds are averaged over the whole domain and the currents mostly have an impact on wind mesoscale anomalies.

No specific tuning has been carried out regarding the simulated wind performances. The PBL height computed by the ABL1D model is ~30% higher in ABL simulations than in FRC and thus induces stronger vertical mixing and surface winds. Hence, it would be possible to achieve a better agreement with observations by tuning the several parameters from the TKE vertical

diffusion scheme as large uncertainties remain regarding these values (Cuxart et al., 2000; Cheng et al., 2002). Illustrations of this sensitivity can be found in the idealized test cases presented in Lemarié et al. 2020.

Assumption is made here that the main source of ageostrophy is through surface turbulent processes. Under such hypothesis, geostrophic winds computed from ERA-interim should be similar to the total ERA-interim winds at the top of the PBL, which is not always the case as we found geostrophic winds stronger than total winds at the top of the PBL on average (not shown).

**3.2 Current feedback in coupled and forced modes**

ABL1D ability to simulate air-sea interactions at mesoscale can be evaluated by computing various coupling coefficients proposed in previous studies (e.g: Bryan et al., 2010; Chelton et al., 2007; Renault et al., 2016b).

Regarding dynamical feedbacks, the coupling coefficients S$\tau$ and Sw (Renault et al., 2016b) are the coefficients of the linear regressions between curl anomalies of the surface current and the surface wind stress and between curl anomalies of the surface

current and the 10m wind speed, respectively. The mesoscale anomalies are isolated by filtering spatially the large-scale structures of the fields. For a field $\Phi$, mesoscale $\Phi'$ anomalies are defined as $\Phi'= \Phi - \Phi_{LS}$ where $\Phi_{LS}$ is the large-scale component of the field. The large scale component signal $\Phi_{LS}$ is obtained as in previous studies (Chelton et al., 2007; Oerder et al., 2016; Renault et al., 2016b) from the field $\Phi$ by applying a high-pass Gaussian spatial filter on $\Phi$. The spatial window area considered for the filter ($6\sigma + 1 * 6\sigma + 1$ with $1° * 1° \sigma$) corresponds to a cut-off of 150km and is used to remove the

large-scale signal from the wind/stress and current fields. Similar to Chelton et al. (2007), the weather-related variability is removed by applying a running monthly average with a 29-days window. Finally, the statistical relationship between surface wind (stress) and surface current curls is then evaluated with binned scatterplots. The slope of the linear regression between surface current and surface wind (stress) curls corresponds to the coupling coefficients Sw (S$\tau$).

Most studies on CFB (e.g: Jullien et al., 2020; Renault et al., 2016b, 2017b) use the geostrophic Sw and S$\tau$ (i.e: the slope of

the linear regression between of the curl of the geostrophic currents and the curl of the surface wind/stress). However, no significant differences are found between geostrophic S$\tau$ and Sw and the total S$\tau$ and Sw in our simulations. Hence, the coupling coefficients here are directly compared to the geostrophic S$\tau$ and Sw in the literature. The values of the coupling coefficients computed in the different experiments are summarized in Table 2.






| Period | $S\tau$ | | | $Sw$ | | | CFB |
|---|---|---|---|---|---|---|---|
| | 2016 – 2017 period | Summer (JJA) | Winter (DJF) | 2016 – 2017 period | Winter (DJF) | Summer (JJA) | |
| FRC_ABS | ~0 | ~0 | ~0 | ~0 | ~0 | ~0 | No |
| FRC_REL | -0.027 | -0.019 | -0.035 | ~0 | ~0 | ~0 | CFB_stress |
| ABL_ABS | ~0 | ~0 | ~0 | ~0 | ~0 | ~0 | No |
| ABL_REL | -0.017 | -0.012 | -0.022 | 0.333 | 0.345 | 0.338 | CFB_stress + CFB_wind |

**Table 2: Dynamical coupling coefficients obtained for all simulations, averaged by season or for the all the 2016 – 2017 period of simulation and over the whole IBI area.**

In FRC_ABS and ABL_ABS simulations, $S\tau$ is close to zero because the currents are not taken into account in the surface stress. When currents are considered in the surface stress (FRC_REL and ABL_REL), $S\tau$ becomes negative (-0.027 N.s.m$^{-3}$ for FRC_REL and -0.017 N.s.m$^{-3}$ for ABL_REL) because the curls of surface stress and current anomalies are opposed.

In FRC_REL and FRC_ABS, $Sw$ is equal to 0 as the winds does not respond to the surface stress anomalies.

A positive correlation is obtained between the curls of the surface wind and surface current ($Sw = 0.33$) as the wind curl adjusts
to the surface stress change and accelerates over positive current anomalies (and inversely). This wind response induces a coupling coefficient $S\tau$ less steep in ABL_REL (-0.017 N.s.m$^{-3}$) than in FRC_REL ( -0.027 N.s.m$^{-3}$). Therefore, the negative relationship between surface current and stress is partially damped by CFB_wind. This adjustment mechanism is consistent with those described in the literature (e.g: Bye, 1986; Duhaut and Straub, 2006; Gaube et al., 2014; Renault et al., 2016b; Rooth and Xie, 1992) and provides an advanced validation of the ABL1D model.

In FRC_REL and ABL_REL, $S\tau$ are almost two times stronger in winter (-0.035 N.s.m$^{-3}$ and -0.022 N.s.m$^{-3}$ respectively) than in summer (-0.019 N.s.m$^{-3}$ and -0.012 N.s.m$^{-3}$ respectively) (Table 2). Such seasonal changes can be attributed to the seasonal variability of the large-scale wind. Indeed, we find a linear negative relationship between $S\tau$ and the total wind speed with regression coefficients of -0.0044 N.s$^2$.m$^{-4}$ and -0.0023 N.s$^2$.m$^{-4}$ in FRC_REL and ABL_REL respectively (Fig. 3.). These values are close to estimations which can be found in theoretical studies (Bye, 1986; Duhaut and Straub, 2006; Gaube et al.,
2014; Renault et al., 2017b; Rooth and Xie, 1992) and to numerical estimations deduced from global coupled ocean-atmosphere models (Jullien et al., 2020; Renault et al., 2019). These results reflect again the ability of ABL1D to realistically represent CFB_tot compared to observations and fully coupled Global Circulation Models (GCM).






From a spatial point of view, Sτ is negative almost everywhere, except in areas of very low EKE as the continental shelf for ABL_REL and FRC_REL (Fig. 4). Positive Sτ over shallow seas could be related to the bottom stress which changes the

ocean-atmosphere coupling behaviour or to the intense tidal motions in the area and should be further addressed in future studies. Sτ is more negative in FRC_REL than in ABL_REL and the strongest Sτ values are found in the North-west. In this part of the IBI area, large-scale winds are usually stronger and induces a more negative Sτ.

## 3.3 Current feedback impact on kinetic energy budget

### 3.3.1 Impact on the surface kinetic energy


The zonal and meridional components of the geostrophic current $u_g$ and $v_g$ are computed from the Sea Surface Height (SSH) and the geostrophic kinetic energy $KE_g$ is computed as:

$$KE_g = \frac{1}{2}\left(u_g{}^2 + v_g{}^2\right) \tag{7}$$

The mesoscale geostrophic Eddy Kinetic Energy (EKE$_g$) is obtained by replacing $u_{geo}$ and $v_{geo}$ from Eq. 7 by the mesoscale components of the current $u_g'$ and $v_g'$ computed from the SSH mesoscale anomalies obtained as described in 3.2. The

geostrophic Mean Kinetic Energy (MKE$_g$) is obtained by substracting EKE$_g$ from KE$_g$.

The CFB decreases KE$_g$ through a reduction of the surface wind-work (Dewar and Flierl, 1987; Duhaut and Straub, 2006; Renault et al., 2016b). In particular, the difference between the coupled simulations (ABL_REL-ABL_ABS) gives an estimation of the KE$_g$ reduction around 19%, which is induced by the CFB_tot (Fig. 5.b). The KE$_g$ decrease occurs during an adjustment phase from the simulation start and lasting approximately 3 months.

In terms of magnitude, the KEg difference FRC_REL - FRC_ABS (-31%) is significantly greater than ABL_REL – ABL_ABS (-19%, Fig. 5.b) because of the atmospheric response (i.e: CFB_wind) which partially re-energize the ocean in ABL_REL. Indeed, in absence of any response of the surface winds to the surface stress conditions, as it is the case in FRC_REL, such stronger winds would imply a stronger KE$_g$ reduction as the efficiency of the KEg damping through CFB_stress increases with the wind amplitude (Bye, 1986; Duhaut and Straub, 2006; Renault et al., 2017b; Rooth and Xie, 1992).


As the EKE$_g$ component represents ~80% of the total KE$_g$ over the IBI area, most of the KE$_g$ decrease between ABL_REL and ABL_ABS is related to the "eddy" CFB-induced change (Fig. 5.b,c). The KE$_g$ damping remains stronger at mesoscale (-28% for FRC and - 18% for ABL) but the current feedback also impacts scales larger than 150km since a reduction of the MKE$_g$ component is observed (-9% for FRC and - 5% for ABL).

The Mediterranean-sea and the North Atlantic drift have the strongest levels of KE$_g$ in the IBI area (Fig. 6.a). However, the relative KE$_g$ differences between ABS and REL simulations "FRC_REL - FRC_ABS" and "ABL_REL -– ABL_ABS" are almost quasi-homogeneous over the whole IBI area for both forced and ABL simulations, except over the continental shelf where the difference of KE is near 0 and daily averaged currents very weak (Fig. 6.b.c).




### 3.3.2 KE budget

To understand how the CFB affects the ocean kinetic energy over the whole water column, this section presents a budget of
KE which is computed online the NEMO model. Only ABL_REL and ABL_ABS simulations are considered here. A
derivation of the Kinetic Energy equation can be written as follow:

$$\frac{\partial KE}{\partial t} = ADV + PW + ZDF + HDF + STR \tag{8}$$

Where ADV is the 3D advection of KE, PW is the pressure work term, ZDF and HDF are the vertical and lateral viscous
dissipation and diffusion terms, respectively, and STR is the stress term. We integrate the KE budget over two domains, a

surface layer, $\mathcal{D}_1$, and a "deep layer", $\mathcal{D}_2$, defined by a bathymetry greater than 2000m and a vertical extension of 0-300m for
$\mathcal{D}_1$ and 300-2000m for $\mathcal{D}_2$. These two budgets are further integrated over a 1-year period, from April 2016 to April 2017.
We found that at the boundaries of $\mathcal{D}_1$ and $\mathcal{D}_2$ the advection of KE and pressure as well as the KE horizontal and vertical
diffusive fluxes are negligible. Therefore, the domain-integrated KE trends over a domain $\mathcal{D}_i$ reduce to:

$$ADV|_{\mathcal{D}_i} \approx 0$$

$$ZDF|_{\mathcal{D}_i} = -\int_{\mathcal{D}_i} U_h \frac{\partial}{\partial z}\left(A_v \frac{\partial U_h}{\partial z}\right) dV \approx -\int_{\mathcal{D}_i} A_v \left|\frac{\partial U_h}{\partial z}\right|^2 dV$$

$$HDF|_{\mathcal{D}_i} \approx -\int_{\mathcal{D}_i} U_h.\Delta_h(A_h \Delta_h U_h)\, dV \approx -\int_{\mathcal{D}_i} A_h |\Delta_h U_h|^2\, dV \tag{9}$$

$$PW|_{\mathcal{D}_i} = \int_{\mathcal{D}_i} \frac{1}{\rho_0} U_h.\nabla P\, dV \approx \int_{\mathcal{D}_i} \rho g w\, dV$$

$$STR|_{\mathcal{D}_1} = \int_{S_1} \frac{1}{\rho_0}\tau.U_h\, dS \text{ and } STR|_{\mathcal{D}_2} = 0$$

Where i = 1 or 2, $U_h = (u, v)$ where $u$ and $v$ are the zonal and meridional currents, $P$ is the pressure, $A_v$ and $A_h$ are the turbulent

vertical and horizontal viscosity coefficient for momentum and $\tau$ is the surface wind stress. Note PW reduces to the Potential
energy (PE) to KE conversion term.

In the surface layer, the only source of KE over the IBI area is STR, whereas PW, ZDF, HDF are sinks of KE (Fig. 7.a). The
PW term, which is a sink of KE in the surface layer, becomes the main source of KE in the "deep layer". This shows that the
flow is mostly driven by pressure at depth as the PW term indicates that the PE to KE conversion is the main source of KE in

the deep layer. In this layer, PW is mainly balanced by HDF (Fig. 7.b) while other terms (ZDF and STR) are negligible.
The KE budget is nearly balanced in the surface layer when averaged over the period considered (Fig. 7.a) and Eq. (8) can be
considered as stationary. Note that the KE budget below the surface layer is not balanced and this could be related to a longer
duration of the spinup period at depth. (Fig. 7.b).
The reduction of KE caused by the CFB in ABL_REL is stronger in the surface layer [0 and 300m] than deeper. However, it

is noteworthy that this reduction occurs down to 1500 m depth (Figure 8.).



In the surface layer, the CFB reduces the KE input through STR by approximately 4%. This forces a slowdown of the current in upper-layers and thus a reduction of viscous dissipation terms ZDF and HDF. The KE to PE conversion through PW is also strongly reduced in the surface layer of ABL_REL. PW plays a strong role in maintaining the balance between sources and sinks of KE as it represents 60% of the total reduction of KE sinks (PW + HDF + ZDF) in the surface layer, whereas ZDF and

HDF contributions are much smaller (25% and 15% respectively). At depth, STR vanishes rapidly as a result of viscous dissipation and cannot explain the KE decrease in the deep layer. In the "deep layer", the KE input through PE to KE conversion (PW) is reduced by 7% and the balance is maintained by a reduction of HDF by 5% (Fig 7.d).

At the ocean surface, the CFB induces surface stress curl anomalies opposed to that of the eddy and the associated Ekman pumping/suction tends to attenuate the horizontal pressure gradients and thus the eddies (Gaube et al., 2014; Seo et al., 2016).

Following Gaube et al. (2014), and since we found no significant changes in the background Ekman pumping (not shown), we examine the changes in the Ekman pumping anomalies induced by the local surface current effect on the relative wind, which is computed as follows:

$$\widetilde{W_{ek}} = \frac{\nabla \times \tilde{\tau}}{\rho_0 \, (f + \zeta)} \tag{10}$$

Where f is the Coriolis frequency, $\rho_0$ is the average density of seawater, $\zeta$ is the geostrophic current vorticity and $\tilde{\tau}$ is the mesoscale anomalies of the surface wind stress, obtained as in part 3.2. For ABL_REL, mesoscale anomalies of Sea Surface

Height (SSHa) are positively correlated with the Ekman pumping velocity anomalies whereas no correlation is observed for ABL_ABS (Figure 9.), meaning that anticyclones (cyclones) are associated with upward (downward) anomalies of Ekman pumping velocities.


Above anticyclones - associated with a negative anomaly of pressure at their centre - current/stress interaction induce a cyclonic stress anomaly that will pump cold water upwards, and inversely for a cyclonic eddy. As a result, the CFB tends to reduce the horizontal pressure gradients at mesoscales and thus the geostrophic currents, therefore explaining the KE decrease in the deep

layer, which is significant down to 1500m depth. By modifying the mass distribution through Ekman pumping anomalies, the current feedback, only active near the surface, has an indirect impact on the circulation and this over the whole water column. The KE reduction is observed deeper during winter (not shown). This is coherent with the results shown in Table 2 as the stronger Sτ found in winter imply that the curl anomalies of the surface currents associated with eddies induce stronger surface stress curl anomalies (and therefore stronger Ekman pumping velocities). As a consequence, the damping of the pressure

gradient is stronger during winter.





### 3.3 Conclusion

The coupling between the ocean and the atmosphere associated with the surface current, CFB_tot, is composed of an oceanic retroaction loop (CFB_stress) induced by the surface current on the wind-stress and of an atmospheric retroaction loop

(CFB_wind) induced by the wind adjustment to the wind-stress change. A one-dimensional atmospheric boundary layer model is used to unravel the two components CFB_stress and CFB_wind of CFB_tot and to investigate impacts of CFB_tot on the kinetic energy budget in the North-East Atlantic, which is an area of low KE. In that way, ABL1D can be viewed as a practical and efficient tool to study mesoscale ocean-atmosphere dynamical coupling processes.

In agreement with former studies, a partial re-energization of the ocean upper-layers, held by CFB_wind, is active when the

ocean is coupled with ABL1D. The model reproduces a similar relationship between the curls of the current and wind/stress as in theoretical studies or fully coupled models (e.g : Bye, 1986; Duhaut and Straub, 2006; Gaube et al., 2014; Jullien et al., 2020; Renault et al., 2016b, 2017b, 2019; Rooth and Xie, 1992). Moreover, the coupling coefficient $S\tau$ increases with the large-scale wind speed in agreement with previous studies based on satellite observations (Renault et al., 2017b) and full coupled models (Jullien et al., 2020). These results show that representing the vertical wind shear changes in the PBL induced

by surface conditions is a key process for a proper representation of CFB_tot.

The current feedback impact on the KE budget is investigated by integrating the KE trends over a surface layer from the surface down to 300m and over a "deep layer" (300m – 2000m). In the surface layer, the CFB affects the KE budget by decreasing the surface wind work which is mainly balanced by a reduction of the vertical mixing and the pressure work (i.e: the production of Potential Energy, PE). In the deep layer, the pressure work switches as a source of KE which is mainly balanced by the

lateral diffusion term. We show that the CFB induce a reduction of the pressure work (i.e: the PE to KE conversion) in the deep layer associated with a reduction of KE which is significant down to 1500 m. This reduction can be explained by the CFB-induced Ekman pumping above eddies: as the polarity of the surface stress curl is opposed to the eddy vorticity, the CFB induces an Ekman pumping that tends to attenuate the eddies and thus the horizontal pressure gradients (Gaube et al., 2014; Seo et al., 2016). Through geostrophic balance, the reduction of the horizontal pressure gradients induces a decrease of the

geostrophic currents and this over the whole water column.

Despite several limitations related to unresolved PBL processes in the ABL1D model (SST-induced changes in the cloud physics, unrepresented advective horizontal and vertical processes and SST-induced pressure gradient modifications), the ABL1D model is a powerful tool for oceanography as its small computational cost (only 12% more than a forced simulation) allows a representation of key turbulent processes within the PBL at the same horizontal and temporal resolution as in the

ocean model. Several features of the CFB remain to be investigated in the IBI area. In particular, intense tidal motions on the continental shelf might have strong implications on the behaviour of the CFB in the area and will be investigated with the ABL1D model in a future study.
**Code and data availability**

The IBI36 configuration used here is based on the NEMO 3.6 version which is freely available at https://www.nemo-ocean.eu/. Model initialisation and boundary conditions are available on demand since it represent hundreds of Go of data. Atmospheric forcing are computed from 6-hour ERA-interim data freely distributed by the European Centre for Medium-range Weather Forecast (ECMWF) at https://apps.ecmwf.int/datasets/data/interim-full-daily/levtype=sfc/. The NEMO 3.6 code modified to include the ABL1D model is available as well as the NEMO namelists used for each simulation and the python code used for

filtering the large scale from the currents and the winds at https://atlas.mercator-ocean.fr/s/oXboWZqYs4NJxDc. Model outputs represent several To of data and are available on demand. ERA-interim data is available. ASCAT winds data and documentation is available are freely distributed by IFREMER on the IFREMER / CERSAT site (ftp://ftp.ifremer.fr/ifremer/cersat/products/gridded/MWF/L3/ASCAT/Daily/).

**Author contribution**

T. Brivoal has written the paper with the help of all the co-authors. G. Samson contributed to the work supervision, H. Giordani helped me to compute and analyse the online KE budget, R. Bourdallé-Badie contributed to the experimental design. F. Lemarié, G. Samson and G. Madec designed and developed a preliminary version of the ABL1d model within NEMO 3.6 stable version.

**Competing interest**


The authors declare that they have no conflict of interest

**Acknowledgments**

Florian Lemarié acknowledges support by the Copernicus Marine Environment Monitoring Service (CMEMS) through contract 22-GLO-HR – Lot 2 (High-resolution ocean, waves, atmosphere interaction)

Florian Lemarié and Gurvan Madec acknowledges by the European Union's Horizon 2020 research and innovation programme under grant agreement No 821926 (IMMERSE).






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



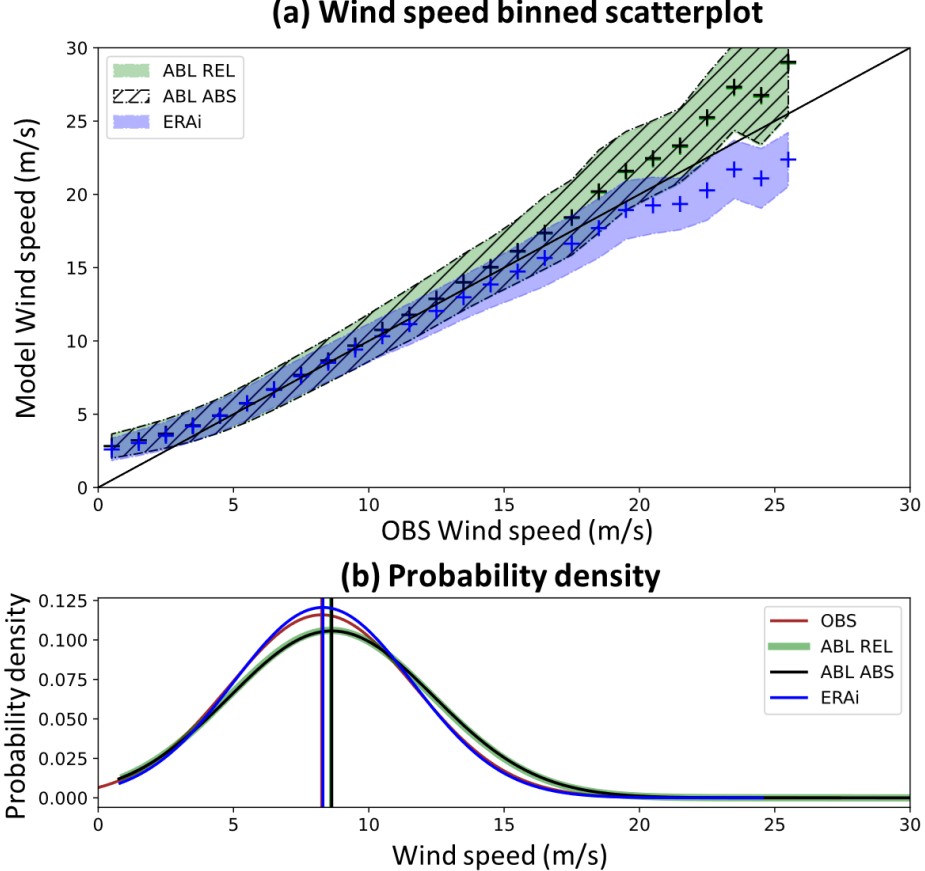

Figure 1: (a) Binned scatterplot of 10-meters stress-equivalent wind (m/s) for ERAI (blue), ABL REL (green) and ABL ABS (black) simulations compared to ASCAT. The crosses represent the binned average and the shaded contours represent ±1 RMSE for ABL REL (green) and ERAI (blue). Black crosses and dashed contours represent the binned average and ±1 RMSE for ABL ABS respectively. The black solid line indicates a perfect fit compared to ASCAT. (b) 10-meters stress-equivalent wind normalised distribution histogram for ERAI (blue), ABL REL (green) and ABL ABS (black) and ASCAT (dark red). Note that the binned averages and wind distribution are almost identical for ABL REL and ABL ABS.





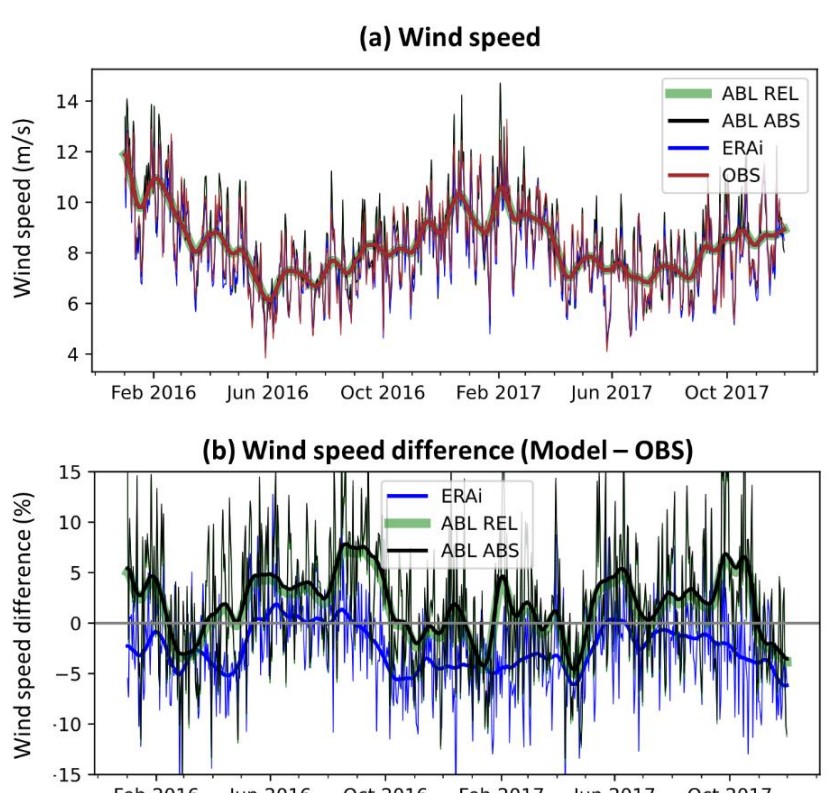

**Figure 2: Time series of (a) stress-equivalent wind speed for ASCAT winds (brown), ABL REL (green), ABL ABS (black) and ERA-interim (Blue) averaged over the IBI area, (b) stress-equivalent wind speed differences in % with ASCAT.**





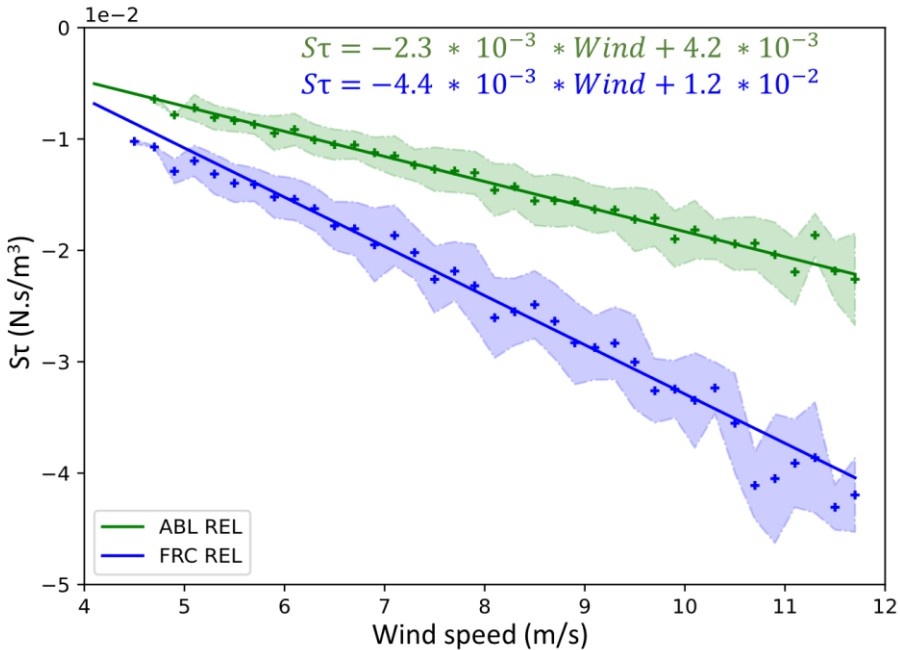

**Figure 3: Binned scatted plots of the coupling coefficient Sτ versus the total wind speed for ABL REL (green) and FRC REL (blue).**
**The dots represent the binned average, the solid line represents the regression line between Sτ and the wind speed and the shaded**
**contours represent ±1 RMSE.**







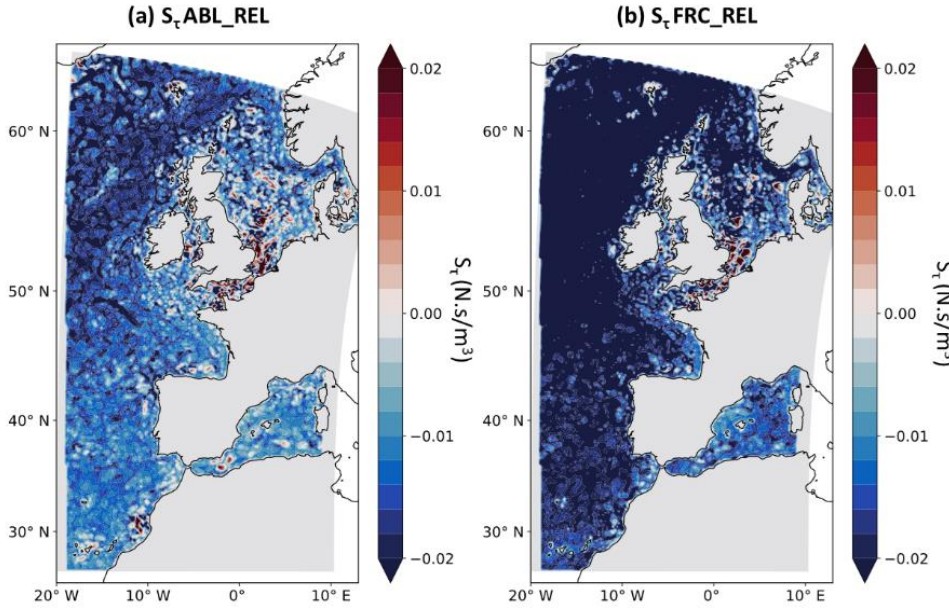

**Figure 4: Maps of the mean coupling coefficient Sτ between the curl of the surface wind stress and the current for ABL REL (a) and (b) FRC REL**

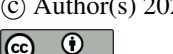

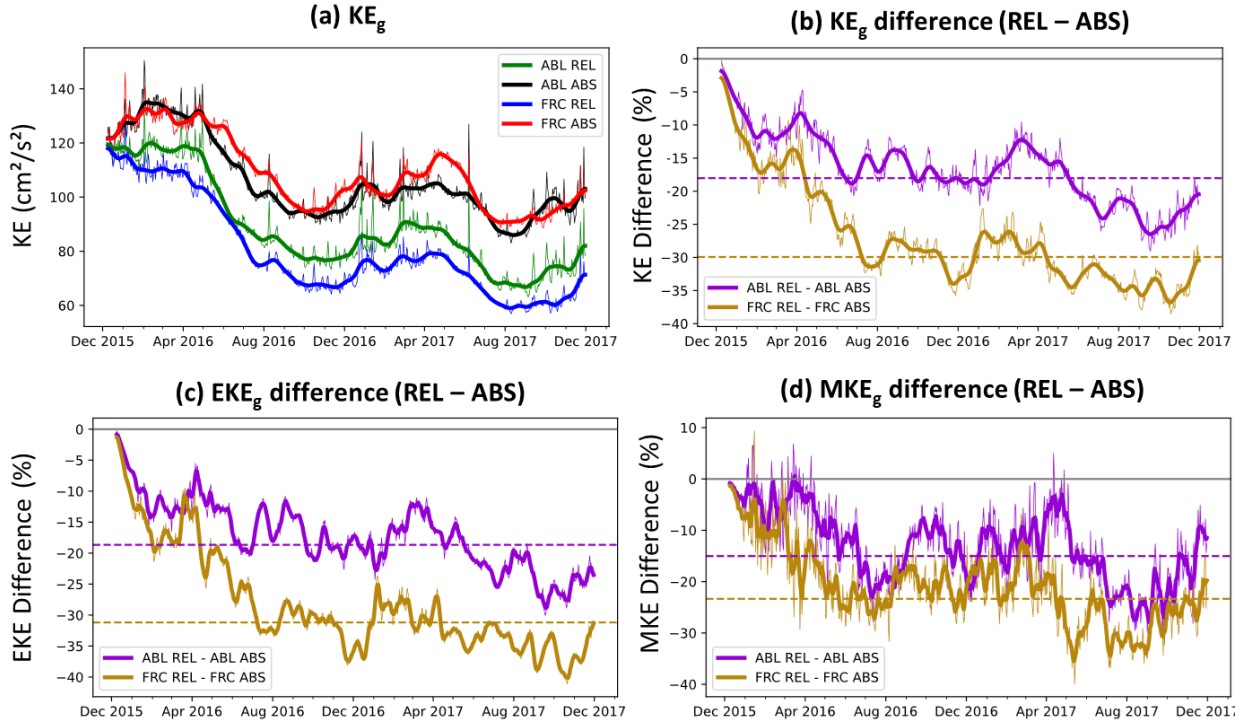

**Figure 5: Time series of the daily means of (a) KE$_g$ for ABL ABS (black), ABL REL (green), FRC REL (blue) and FRC ABS (red) and (b) (c) (d): respectively, KE$_g$, EKE$_g$ and MKE$_g$ differences in % between ABL REL and ABL ABS (violet) and between FRC REL and FRC ABS (golden) averaged over the IBI area. The dashed line represents the mean obtained by removing the spinup period (i.e: the first 3 months) from the data.**




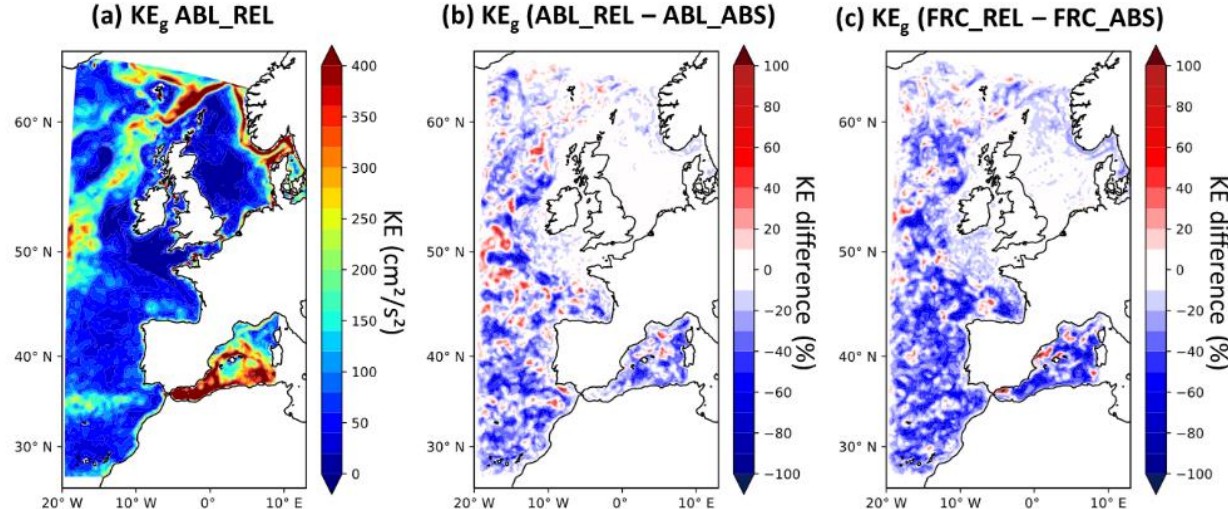

**Figure 6: Map of (a) KE$_g$ mean for ABL REL in cm²/s², (b) KE$_g$ mean difference between ABL REL and ABL ABS in %, and (c) KE$_g$ mean difference between FRC REL and FRC ABS averaged over 2016-2017.**







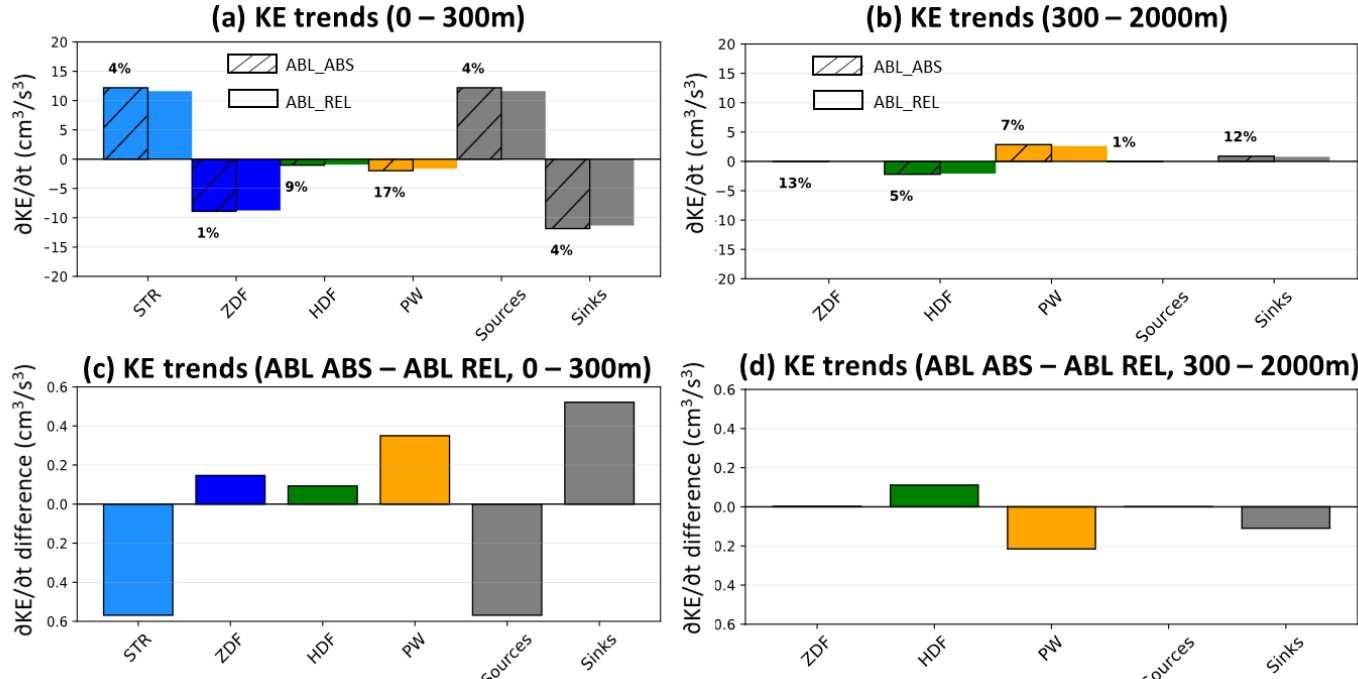

**Figure 7: Mean depth-integrated KE trends over the IBI area for the whole period of simulation in cm3/s3 (a) and difference between the mean depth-integrated KE trends in cm3/s3 from ABL ABS and ABL REL (b) averaged over the 04/2016 – 04/2017 period. The percentage in (a) represents the percentage of change of each trend between ABL ABS and ABL REL, a positive value indicates that the trends are stronger for ABL ABS. The two rightmost columns represent the sum of the source terms in the KE budget (i.e: positive values) and of the sink terms (i.e: negative values).**

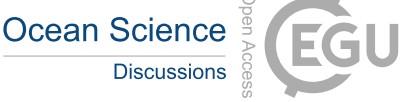



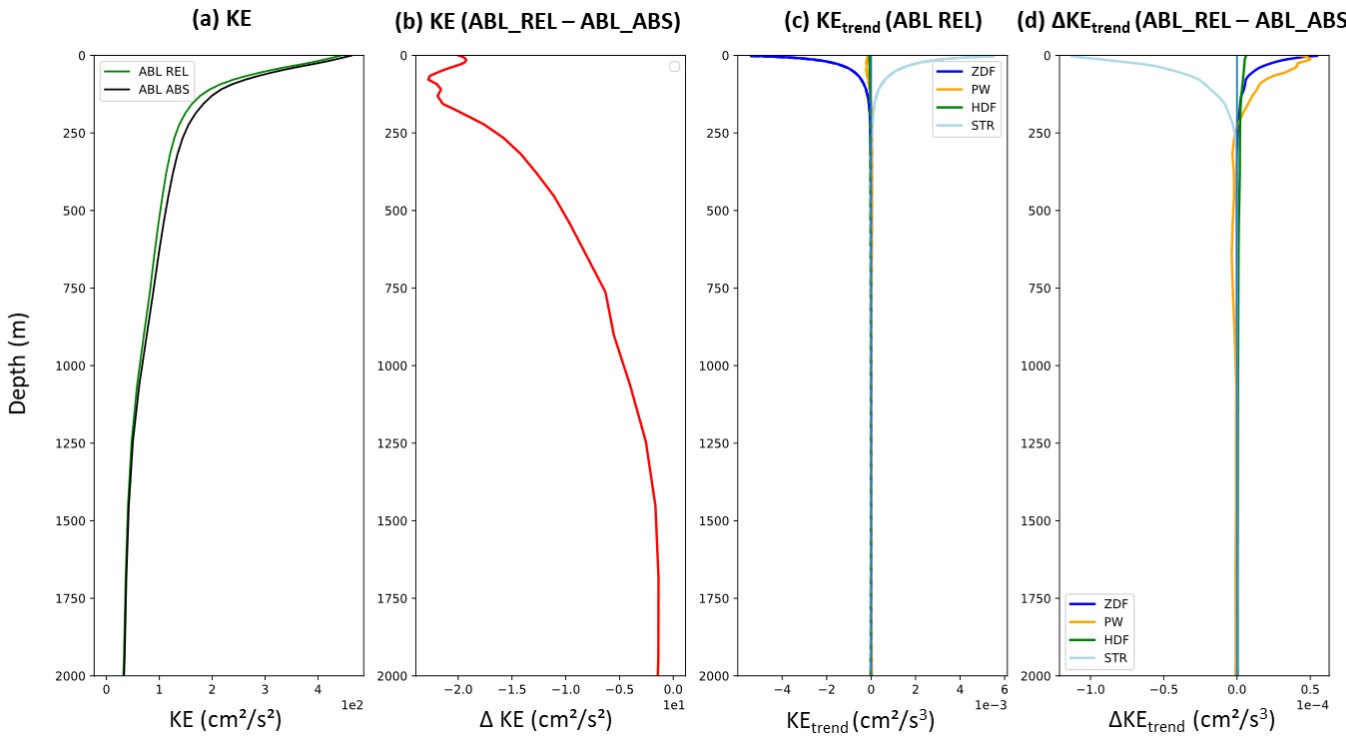

**Figure 8: Mean vertical profile averaged over the IBI area and over the 04/2016 – 04/2017 period for (a) the KE in cm²/s², for ABL REL (green) and ABL ABS (black), (b) the differences of KE between ABL REL and ABL ABS in cm²/s², (c) each KE trends in m²/s³, (d) the differences of each KE trends between ABL ABS and ABL REL in cm²/s³.**



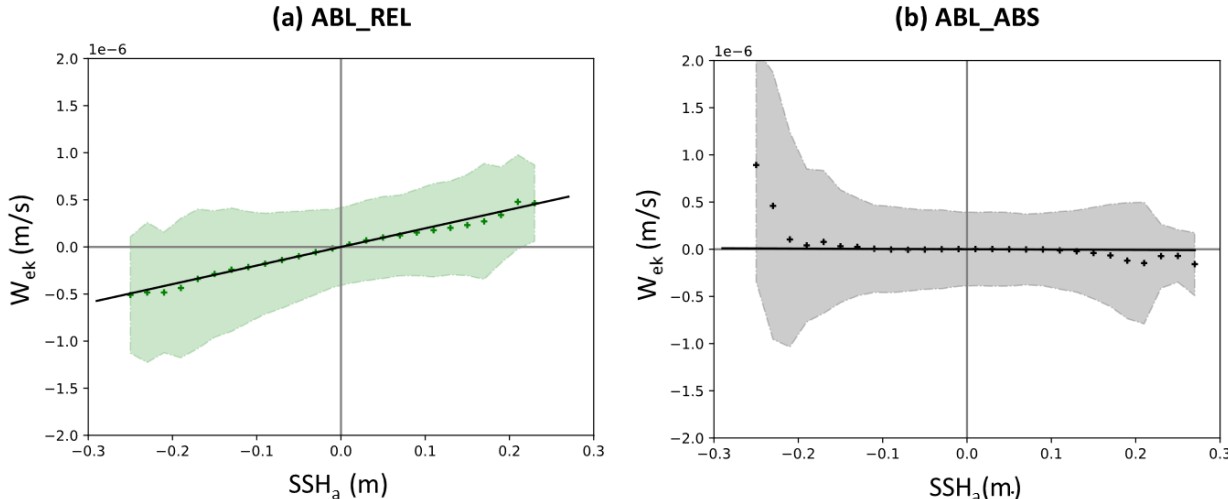

**Figure 9: Binned scatterplots of the mesoscale anomaly of Ekman pumping velocity (m/s) versus the Sea Surface Height mesoscale anomaly (m) for (a) ABL REL and (b) ABL ABS. The dots represent the binned average, the solid line represents the regression line between Sτ and the wind speed and the shaded contours represent ±1 RMSE.**

645