# Peer review of "Impact of the current feedback on kinetic energy over the North-East Atlantic from a coupled ocean / atmospheric boundary layer model."

_Ocean Science, 2020_

## Referee Comment (RC1) · Christopher W. Hughes (Referee) · 17 Mar 2021

The authors have performed a very interesting set of experiments which have the potential to elucidate both the performance of a 1D atmospheric boundary layer model and the mechanisms by which the ocean mesoscale influences atmosphere-ocean momentum fluxes. There are (at least) four different issues involved here: 1) Accounting for ocean currents reduces the energy input to the ocean (many previous studies) 2) Allowing an atmospheric response to the ocean mesoscale reduces this effect (the Renault and Jullien papers cited by the author) 3) Sea surface temperature also influences the wind stresses (e.g. O'Neill et al.). 4) There are two mechanisms for the

SST effect. (Incidentally, one thing I wasn't sure about is whether the ABL model is expected to be able to simulate both SST effects. It would be nice to have a mention of this somewhere.)

The diagnostics presented seem to confirm that effects 1) and 2) occur in this model, as well as showing that the resulting changes in mesoscale energy result from changes in the pressure work in the model. This is quite nice in that it demonstrates that the ABL model can reproduce the mitigating effect 2), seen in more complete atmospheric models. However, the paper does little to pick apart how the ABL model is producing these effects, whether the SST effects are playing a role, and whether the ABL model is capturing these SST effects. That is a pity given the experimental design. In fact the (now published) Lemarie et al. paper describing the 1D model does more along these lines, showing that positive correlations between winds and mesoscale SST arise in the more energetic midlatitude regions.

Overall, I feel there is a lack of focus to the paper. It presents a range of diagnostics, but often doesn't give the context or make clear what is being learnt from them. What is really needed is to be clear about what is being added and elucidated here, and what is reiteration of established results. There are some useful new results, and a little more work would bring more to light. Some reorganisation and better signposting to the reader of the most significant results would be very helpful, as would some additional diagnostics and discussion.

Of the latter, I would particularly like to see an expansion of the coupling coefficient data shown in Figure 3, and more discussion of the relationships found. These are summarised as "consistent with the values in the literature", but there is a lot more to be said than that. For example: 1) Is the difference between the two curves entirely due to a damping of the current feedback by boundary layer dynamics, or is the SST effect also playing a role? The latter could be judged by plotting the results for ABL ABS to see whether the SST effect alone makes a difference (and FRC ABS for completeness) 2) Renault et al. (2017) derive a theoretical relationship for this coupling coefficient

ignoring feedbacks: S_tau = -1.5rho.Cd*Wind, which is -2.2e-3*Wind for their Cd=1.2e-3 and rho=1.225. The slope (if not the intercept!) of their relationship is quite close to what they find in observations. The slope here is twice as steep in the matching FRC REL case. Why could that be? What value of Cd is used in these simulations (if Cd depends on winds, then what is the range over these wind speeds)? 3) How about coupling constants for SST? O'Neill et al. calculated these from observations, so how does this model compare with those data? You have the data here to calculate these coefficients both with and without the current feedback, to see how these interact.

Similarly, when it comes to Figure 5, there seems to be no discussion of 5a, and all the focus is on the REL-ABS differences. While this is interesting in that it illustrates the mitigation of this KE reduction when the ABL is used, it again doesn't address the SST effect which should be shown by the difference between ABL ABS and FRC ABS. While this is not as consistent as the ABL REL minus FRC REL difference, it is usually the same sign and comparable in size, suggesting that the SST effect is playing a role in the mitigation of energy reduction, it isn't just the ABL response to ocean currents.

One final general point - I felt rather saturated in TLAs (Three Letter Acronyms) as I read the paper. I appreciate that it would get rather long-winded to write everything out in full, but using the full terms sometimes would offer some respite to the reader.

There are a number of more technical minor issues that I would like to see addressed, listed below, but the main thing that would greatly strengthen the paper is a clear dissection what is being learnt about the different effects discussed above, giving a better focus in the purpose of the paper, particularly in the context of what has already been presented by Lemarie et al.

Minor Issues:

1) Line 20 "induces"

2) Line 21-22 "and this" acts? "over the whole water column"

3) Line 42 "sets up" -> is set up

4) Line 44 "consider" -> considering

5) Line 48 (or perhaps earlier) - wind-work here refers to the work on the geostrophic part of the flow. The total work on the ocean is much larger, but much of it is dissipated immediately in the surface mixed layer. Somewhere this distinction should be made clear.

6) Line 55 - eddies -> eddies'

7) Section 2 - begin by explaining that the (now published) Lemarie et al. paper gives the complete description and first validation results for this model.

8) Line 82 - "models" -> model

9) Lines 103-118 - this is rather a confusing description. It seems to say that geostrophic winds are derived from MSLP, which is calculated as a combination of u,v,theta,q and MSLP. The description in Lemarie et al. seems clearer, and doesn't have a geostrophic U in their version of Eq. 1, but $R_{LS}$ - a geostrophic plus relaxation term. Presumably that is where the other variables come in? And it would be helpful to specify whether this term is independent of height.

10) Lines 134-135 "drag coefficient-induced SST changes" - do you mean SST-induced drag coefficient changes? SST doesn't change in response to changes in drag coefficient, but the drag coefficient does change in response to SST.

11) Line 139 - "total kinetic energy" per unit mass

12) Lines 139 to 141 - $C_s$, $C_m$ and $C_1$ occur in the equations, but only $C_m$ is given a value (4). Is $C_e$ supposed to be $C_m$? And is $C_1$ something else?

13) Line 210 "were" -> was

14) Line 226 - I couldn't see why the conversion to equivalent neutral winds was done

here, why not stick with stresses?

15) Line 228 "1,22" -> 1.22

16) Line 252 - An assumption is made here...

17) Line 256 - The ability of ABL1D to simulate...

18) Line 263 - low-pass, not high

19) Line 263-265 - This is not clear. Is sigma 1 degree? In latitude and longitude, or great circle? Is sigma the standard deviation of the Gaussian used? What is meant by the 1 in 6sigma+1, 1 degree?

20) Table 2 - give units for S_tau.

21) Section 3.3.2 - There's also the d(KE)/dt term, which is easily calculated as H*[KE(end)-KE(start)]/time where H is layer thickness. This comes out at around 0.04 for the 300 m surface layer, and about 0.2 for the 1700 m deeper layer using values from Fig. 5a (it is unclear what these values are - domain averaged? over all depths?), so there's no need to speculate about longer spinup period at depth, the values can simply be added to the budget if diagnosed over the relevant volumes. There also seem to be sign errors in Eq. 9. The middle terms of the 2nd and 3rd lines, and the final term of the 4th should be the opposite sign.

22) Lines 360-360 - I don't understand what this is showing. If STR is the energy input by wind stress at the surface it should be zero everywhere below the surface. It seems to be balanced by the vertical viscosity term - I think explicit formulae for the terms being shown are needed rather than just these verbal descriptions.

23) Eq. 10 - if zeta is to be accounted for here, it should be inside the curl operator (as, strictly, should f) - the right hand side should be (1/rho)curl(tau/[f+zeta]).

24) Lines 370-380: This needs to be clarified. Upward Ekman pumping is associated with anticyclones (sea level high, and positive, not negative pressure anomaly). This

represents damping because these baroclinic eddies are a sea level high balanced by a dip in the thermocline. The Ekman pumping damps out this dip in the thermocline. Alternatively, the Ekman flux divergence is a flow down the horizontal pressure gradient, and thus a loss of energy from the eddy.

25) Lines 397-8 - as discussed above, this point needs much more discussion and quantification.

26) Line 404 - "switches" -> acts

27) Line 410 - "and this" occurs "over the whole water column".

28 "Code and data availability" "Go" and "To" -> Gb and Tb.

---

## Referee Comment (RC2) · Anonymous Referee #2 · 24 Mar 2021

The authors ran four numerical experiments on the effect of ocean current feedback; with and without the one-dimensional atmospheric boundary layer model; using atmospheric winds absolute and relative to ocean currents. The OGCM was sub-mesoscale resolving (1/36 deg). In terms of wind speed, the effect of ocean current was small ($<$ 10%), but discernible in coupling coefficients. The resultant coupling coefficients are consistent with previous results. The effect on ocean currents is to reduce the kinetic energy, which was significant. The effect can be seen below 1500 m depth, which is explained by a change in the Ekman pumping.

The aims of the present work was twofold (L.90) – to validate the atmospheric boundary

layer model and to quantify the ocean current feedback effects.I found problems for both.

For validation of the ABL1D model, it was not clear to me what was the true value. The ASCAT product has an error of 1 to 2 m/s relative to buoy wind speed (Fig.11, Bentamy and Fillon, 2012, `https://doi.org/10.1080/01431161.2011.600348`). This is of a similar magnitude to the RMS difference between ASCAT and ABL (L.234, 1.98 m/s) and to that between ASCAT and ERA-interim (L.236, 1.43 m/s). It is probably valid to assume that ASCAT is the true value, but it was not clear to me if the use of the ABL model improved the wind output and/or ocean state estimate. Fig.1b suggests that ABL model did not change the distribution of wind speed much, which is not validation. In Section 3.2, the model was deemed validated based on the qualitative agreement of the present result with past work, some of which are theoretical and simulations. (L.289, L.297). I am not convinced that the ABL model is validated with these attempts.

For the ocean current feedback, I failed to see the significant of the present work, although many experiments were described in detail. I could not point out what was discovered in this work and what was already known. The results are often described without interpretation (e.g. why the KE response is quasi-homogeneous in Fig.6 despite its inhomogeneous mean state?). I also could not find the reason why the authors chose "a region of low mesoscale activity" (L.90) for this particular study.

Based on these problems, I was not convinced that the manuscript merits publication in the present form. A major revision might be able to clarify the novelty of the present work and to place it in the context, but I feel it more appropriate to handle it as a re-submission.

**L.41,** Small et al.,(2008) not found in the reference list.

**L.69,** "significative" → significant?

**Eq.4,** The curly brackets do not make sense. Also I could not find the definition of C1.

**L.228,** Change comma to period in "1,22"

**L.264,** What is $\sigma$ here?

**Tables,** Captions to tables are usually placed above the table.

---

## Author Comment (AC1) · 19 May 2021

Many thanks for your interest and your comments on the paper. We respond to your main points:

1 – *"The paper does little to pick apart how the ABL model is producing these effects, whether the SST effects are playing a role, and whether the ABL model is capturing these SST effects».*

In our experimental setup, the tracers (i.e : the potential air temperature and the humidity) are fully nudged towards ERA-Interim by using a relaxation time on the tracers equal to one model time step. This means that with this setup, the tracers cannot react to the ocean surface. As a result:

- The SST / wind or SST / stress coupling coefficients (Chelton et al., 2007; O'Neill et al., 2012) are ~0 when the tracers are fully nudged (Figure R1). This means that the thermal feedbacks between the SST and the atmosphere are negligible in our simulations.
- The ocean surface can only alter ABL1D winds through changes in the surface stress.
- Since the SST / stress coupling coefficients are negligible with this setup, surface stress changes are mostly related to ocean currents.

Moreover, and this is now mentioned in the new version of the paper, the dynamical coupling coefficients ($S\tau$ and $Sw$) are almost unsensitive to the relaxation imposed to the tracers (when the tracers are not fully nudged towards ERA-Interim, Figure R2). This suggests that the currents mostly interact with the winds by changing the surface stress, and therefore the vertical wind shear.

However, we agree with you that all these points were probably not clear enough in the previous version of the paper. We added more clarifications about this in the new version:

- In sec. 2.1 : "With this setup, ABL1D winds are only impacted by the ocean surface through the surface stress (and not by the turbulent heat fluxes). Therefore, this setup allows us to efficiently isolate the effect of the current feedback from other coupling processes. Note that this is different from Lemarié et al. (2020) where the tracers are modified by the ABL1D."

- In sec 3.2 : «As mentioned in sec 2.1, the tracers in ABL REL or ABL ABS are not impacted by the ocean's surface since they are fully nudged towards ERA-Interim. This means that in ABL REL and ABL ABS, the ocean surface can only have an impact on winds through changes in surface stress. Moreover, SST / wind coupling coefficients (Chelton et al., 2007; O'Neill et al., 2012) are near 0 in all simulations (not shown) and thus indicates that the SST / wind or SST / stress coupling is not present. Therefore, differences between $S\tau$ of ABL_REL and FRC_REL are not attributable to the impact of SST on winds. We also find that $S\tau$ and $Sw$ are almost unsensitive to the relaxation imposed to the tracers (not shown). This suggests that the vertical wind shear adjustment to surface stress is the main driver of CFB_tot since it is the only mechanism that could explain the positive $Sw$ found in ABL REL. »

This also respond to the question: « Is the difference between the two curves entirely due to a damping of the current feedback by boundary layer dynamics, or is the SST effect also playing a role? ». Since

the SST / wind feedbacks is not present in our simulations, the KE differences between ABL_REL and ABL_ABS are only due to the current feedback.

However, differences between FRC_ABS and ABL_ABS are also related to differences in background winds. We also added clarifications on this point in the paper in sec 3.3 : « $KE_g$ in ABL ABS and FRC ABS are in the same order of magnitude (Fig 5.a), differences between the two simulations are only attributable to the changes in background winds between the two simulations as the SST feedback to the atmosphere is not present in both simulations. »

*2 – « What is really needed is to be clear about what is being added and elucidated here, and what is reiteration of established results. There are some useful new results, and a little more work would bring more to light. Some reorganisation and better signposting to the reader of the most significant results would be very helpful, as would some additional diagnostics and discussion. »*

We reorganised and added discussions in the paper to clarify what is new from our study and what was already known.

Major changes :
- Fig 5 and 6 : Total KE is used rather than just geostrophic KE only to be consistent with the rest of the paper.
- We reorganised section 3: 3.3.1 becomes section 3.3 : *Impact on kinetic energy* and section 3.3.2 becomes section 3.4 : *Current feedback impact on kinetic energy budget over the water column*
- Figure 8 is now Figure 7, and now represents only the impact of the current feedback on KE, since the vertical profile of trends did not provide new information. We also added a vertical profile of KE differences between ABL REL and ABL ABS in % of ABL ABS.
- Discussions added in sections:
  - Introduction : clarifications on "why we chose a region of low mesoscale activity"
  - 3.1 : discussion about the differences between ERAi, ASCAT and ABL1D winds and what can cause these differences.
  - 3.2 : discussions about how the ABL1D model is producing the wind response to the surface currents, and about the values of the slope between $S\tau$ and the background winds.
  - 3.3 : discussions added on why the KE response is quasi-homogeneous despite its inhomogeneous mean state and more, and on how the ABL1D is producing the KE partial-reenergisation. We also added discussion about the KE differences between ABL REL and ABL ABS at depth: to our knowledge, this is the first time a study shows that the current feedback can have an impact on the ocean at such depth (1500m). We also added signposting to introduce part 3.4.
  - 3.4 : Clarifications about Ekman pumping mechanism added
  - Conclusion & abstract: we clarified what is new in our study.

- Clarifications about the nudging of the tracers have been added in section 2.1.

*3 – « Renault et al. (2017) derive a theoretical relationship for this coupling coefficient C2 ignoring feedbacks: S_tau = -1.5rho.Cd\*Wind, which is -2.2e-3\*Wind for their Cd=1.2e3 and rho=1.225. The slope (if not the intercept!) of their relationship is quite close to what they find in observations. The slope here is twice as steep in the matching FRC REL case. Why could that be? What value of Cd is used in these simulations (if Cd depends on winds, then what is the range over these wind speeds)  »*

This is an interesting point. The slope we found for ABL_REL (-2.3e-3*Wind) is quite close to the slope they found in observations (-2.5e-3*Wind) and from the slope found in Jullien et al. (2020) from a fully coupled ocean-atmosphere model (-2.3e-3*Wind). However, as you mentioned the analytical relationship derived in Renault et al. (2017) should be representative of the relationship found in FRC_REL since feedbacks are ignored in the analytical formulation. If we compute the slope from their relation from the Cd in FRC_REL and assuming rho=1.225, we found a slope of $-2.7^e$-3*Wind which is still quite far from the slope in FRC_REL. The reasons for such a discrepancy are not clear, this might be related to the filtering technique or to the approximations made in the Renault et al (2017) formulation but should definely be addressed in further studies. Nonetheless, since we use a similar filtering technique as in Renault et al. (2017) or Jullien et al. (2020), the slope found in ABL_REL can be compared to the slope computed from observations in Renault et al. (2017) or coupled models in Jullien et al. (2020).
We added a discussion for this point in the paper in sec 3.2.

4 – Minor technical issues have been adressed in the new version of the paper.

- *« Lines 103-118 - this is rather a confusing description. It seems to say that geostrophic winds are derived from MSLP, which is calculated as a combination of u,v,theta,q and MSLP. The description in Lemarie et al. seems clearer, and doesn't have a geostrophic U in their version of Eq. 1, but R_LS - a geostrophic plus relaxation term. Presumably that is where the other variables come in? And it would be helpful to specify whether this term is independent of height. »*

When the geostrophic winds are used, there is no relaxation done on the dynamics since the model is guided by the large-scale pressure gradients (fk.Ugeo). However, this could be the case when the equatorial region is considered. We added clarifications in the paper.

- *« Line 226 - I couldn't see why the conversion to equivalent neutral winds was done C4 here, why not stick with stresses? »*

ASCAT scatterometers directly measure equivalent neutral winds (ENW), so it was cleaner to directly compare ENW rather than stresses.

- *« Eq. 10 - if zeta is to be accounted for here, it should be inside the curl operator (as, strictly, should f) - the right hand side should be (1/rho)curl(tau/[f+zeta]). »*

We followed Gaube et al. 2015, in which Ekman pumping was computed this way. However, it does not change much the results if [f+zeta] are accounted into the curl operator.

[Figure]

Figure R1 : Mean thermal coupling coefficients (averaged over the IBI area) computed from 1/12° NEMO / ABL1D simulations performed over the year 2017 against relaxation time imposed on the tracers in ABL1D model for a) SCwind = crosswind SST gradient anomaly vs the wind curl anomaly, SDwind = Downwind SST gradient anomaly vs wind speed divergence anomaly, b) SCwind = crosswind SST gradient anomaly vs the stress curl anomaly, SDwind = Downwind SST gradient anomaly vs stress speed divergence anomaly, c) Su = SST anomaly vs wind speed module anomaly and d) Sstr = SST anomaly vs wind stress anomaly.

[Figure]

Figure R2 : binned scatterplots between a) wind stress curl anomaly and the current curl anomaly and b) wind curl anomaly and the current curl anomaly for 2 simulations of one year (2017) at 1/12° resolution over the IBI area : ABL TRC5 REL (black) using relative winds and a (weak) relaxation of 5% (2,5h) on the tracers and ABL TRC100 REL (black) using relative winds and a relaxation of 100% (1 time step) on the tracers (as it is in the paper). The slope of a) corresponds to the $S\tau$ coupling coefficient and b) Sw.

---

## Author Comment (AC2) · 19 May 2021

Thank you for your remarks. Here are our responses for your main points:

1 – *"For validation of the ABL1D model, it was not clear to me what was the true value. The ASCAT product has an error of 1 to 2 m/s relative to buoy wind speed (Fig.11, Bentamy and Fillon, 2012, https://doi.org/10.1080/01431161.2011.600348). This is of a similar magnitude to the RMS difference between ASCAT and ABL (L.234, 1.98 m/s) and to that between ASCAT and ERA-interim (L.236, 1.43 m/s). It is probably valid to assume that ASCAT is the true value, but it was not clear to me if the use of the ABL model improved the wind output and/or ocean state estimate. Fig.1b suggests that ABL model did not change the distribution of wind speed much, which is not validation. In Section 3.2, the model was deemed validated based on the qualitative agreement of the present result with past work, some of which are theoretical and simulations. (L.289, L.297). I am not convinced that the ABL model is validated with these attempts."*

The ABL1D model is strongly simplified and is likely to compute less realistic winds than ERA-Interim, in which atmospheric data is assimilated. However, the aim of this part of the paper was to show that winds computed by the ABL1D model are quite close to ASCAT winds, despite the simplicity of our model (and, therefore its very low computational cost). We clarified this point in the new version of the paper.

On the same manner, we showed that ABL1D model is able to reproduce coupling coefficients consistent with previous results from ocean – atmosphere coupling, and therefore the wind response to the currents. Clarifications have also been added in the new version of the paper.

2 – *"For the ocean current feedback, I failed to see the significant of the present work, al-though many experiments were described in detail. I could not point out what was discovered in this work and what was already known. The results are often described without interpretation (e.g. why the KE response is quasi-homogeneous in Fig.6 de-spite its inhomogeneous mean state?). I also could not find the reason why the authors chose "a region of low mesoscale activity" (L.90) for this particular study."*

We added discussions and reorganised the paper to clarify what is already known and what is new from our study (capability of a super-simplified model to represent major part of dynamical ocean/atmosphere interaction, impact of current feedback at depth).
Major changes :
   - Fig 5 and 6 : Total KE is used rather than just geostrophic KE only to be consistent with the rest of the paper.
   - We reorganised section 3: 3.3.1 becomes section 3.3 : *Impact on kinetic energy* and section 3.3.2 becomes section 3.4 : *Current feedback impact on kinetic energy budget over the water column*
   - Figure 8 is now Figure 7, and now represents only the impact of the current feedback on KE, since the vertical profile of trends did not provide new information. We also added a vertical profile of KE differences between ABL REL and ABL ABS in % of ABL ABS.
   - Discussions added in sections:
      o Introduction : clarifications on "why we chose a region of low mesoscale activity"
      o 3.1 : discussion about the differences between ERAi, ASCAT and ABL1D winds and what can cause these differences.

- o   3.2 : discussions about how the ABL1D model is producing the wind response to the surface currents, and about the values of the slope between $S_\tau$ and the background winds.
- o   3.3 : discussions added on why the KE response is quasi-homogeneous despite its inhomogeneous mean state and more, and on how the ABL1D is producing the KE partial-reenergisation. We also added discussion about the KE differences between ABL REL and ABL ABS at depth: to our knowledge, this is the first time a study shows that the current feedback can have an impact on the ocean at such depth (1500m). We also added signposting to introduce part 3.4.
- o   3.4 : Clarifications about Ekman pumping mechanism added
- o   Conclusion & abstract: we clarified what is new in our study.

- Clarifications about the nudging of the tracers have been added in section 2.1.